# Exploring the Impacts of Urbanization on Eco-Efficiency in China

Xinyue Yuan [1], Yang Nie [2,*], Liangen Zeng [3], Chao Lu [4] and Tingzhang Yang [5]

1    School of Insurance, Central University of Finance and Economics, Beijing 100098, China
2    Guanghua School of Management, Peking University, Beijing 100871, China
3    College of Urban and Environmental Sciences, Peking University, Beijing 100871, China
4    Law School, Shandong University of Technology, ZiBo 255000, China
5    College of Business and Economics, Australian National University, Canberra, ACT 2600, Australia
*    Correspondence: nieyang@pku.edu.cn

**Abstract:** It is of significant importance to conduct research on the relationship between urbanization and eco-efficiency (*EE*), for it can aid policy making for urban and regional sustainable development. This paper studied the effects of urbanization on the *EE* in 30 provinces of China from 2008 to 2019. Using the epsilon-based measure (EBM) model with undesirable outputs, this study measured the *EE* of China's provinces before empirically analyzing the effects of urbanization on *EE*. Conclusions could be drawn: the annual mean *EE* of the eastern region was the highest (0.837), followed by those of the central region (0.653) and western region (0.570), and that of the northeast region remained the lowest (0.438). Zooming into the provinces and cities, the *EE*s of Beijing, Shanghai, and Fujian were at the production frontier surface, with a high level of *EE* during the study period, while those of Gansu, Ningxia, and Xinjiang were generally at a lower level. Empirical analysis showed that the effects of urbanization on *EE* in China presented a U-shaped relationship, having a negative correlation first and then reversing to a positive one. At present, China is in the early negative stage, and the turning point has yet to come. Considering the control variables, the economic development level, technological progress, and foreign direct investment have positively influenced eco-efficiency. Overall, the paper may shed light on related studies and provide relevant policy suggestions to promote *EE* through a new urbanization strategy.

**Keywords:** eco-efficiency; urbanization; influence mechanism



## 1. Introduction

Due to the enormous volume of global resource consumption, ecological problems have become an important issue for a shared future of the world. China is now at an accelerated stage of urbanization, with the rate reaching as high as 58.52% in 2017. It is estimated that by 2030, China's urbanization rate will climb to 70.12% [1], and by 2050, it will exceed 90% [2]. The rapid urbanization in China plays an essential role in improving the Chinese people's living standards and promoting China's social development. However, several recent environmental issues appeared. The large-scale urbanization in China requires magnificent products, energy materials, and other natural resource inputs, resulting in huge waste streams and emissions such as $CO_2$. Since 2011, China has surpassed the United States to become the country with the largest energy consumption and carbon emissions in the world. The accompanying energy and ecological environment problems have seriously constrained the quality of China's social and economic development, as well as its urbanization process. Currently, China's economy has embraced the "new normal", and the traditional linear industrial practice, characterized by high capital investments, massive resource consumption, and severe pollution, could no longer adapt to the domestic situation. Therefore, it is necessary to build up a conservation-minded society to ensure the persistent and healthy development of the economy.

Eco-efficiency (*EE*) can effectively measure the relationship between economy, resources, environment, and development [3]. Understanding eco-efficiency is of great practical significance for decision makers addressing and delivering sustainable socioeconomic development. Thus, improving *EE* is the only path to promoting economic growth coordinated with development quality. It is no wonder that a critical question of whether the current urbanization process in China can improve *EE* has been raised. Therefore, research on the relationship between urbanization and *EE* is of practical significance for understanding and planning the development process of urbanization.

Urbanization brings the accumulation of human resources, capital, and technology, which can trigger higher, cleaner production technology and improvements in energy efficiency. At the same time, however, urbanization is accompanied by the excessive consumption of energy and resources, resulting in pollution emissions [4]. The impacts of urbanization on *EE* are complex, and different phases of urbanization may have different influence on *EE*. As mentioned above, the relationships between urbanization and *EE* have not been clarified. So what will the process of urbanization do to the *EE* in China? Is it good or bad? To answer the questions, this paper intends to analyze the relationship between urbanization and *EE* with an empirical approach.

Compared to what previous scholars have done, the marginal contributions are as follows: firstly, relatively few have explored how urbanization affects *EE*. We focus on this research gap and empirically test the influence of urbanization on *EE* and its mechanism. Secondly, the methods used in the existing literature to measure *EE* mainly included traditional data envelopment analysis (DEA)-based radial and non-radial methods, which have drawbacks that may cause results to deviate from the norm [5]. On the contrary, this paper selected the epsilon-based measure (EBM) DEA model with undesirable outputs to assess the ecological efficiency (*EE*) of 30 provinces in China from 2008 to 2019. Then, the effects of urbanization on *EE* were empirically analyzed by the Tobit model. Other possible factors, including the economic development level, technical progress level, and foreign direct investment were also discussed. Accurately understanding the development of *EE* is the basis of working out a reasonable emission reduction plan, which has the significance of promoting the development of the green economy to make clear the factors affecting *EE*.

The rest of the article is composed of the following parts: Section 2 introduces the data and methodologies. Section 3 presents the characteristics of *EE* in China. In Section 4, the Tobit method is applied to test the influence of urbanization on *EE* in China, and Section 5 summarizes the article and offers suggestions for improvements. The flowchart of the empirical research is shown in Figure 1.

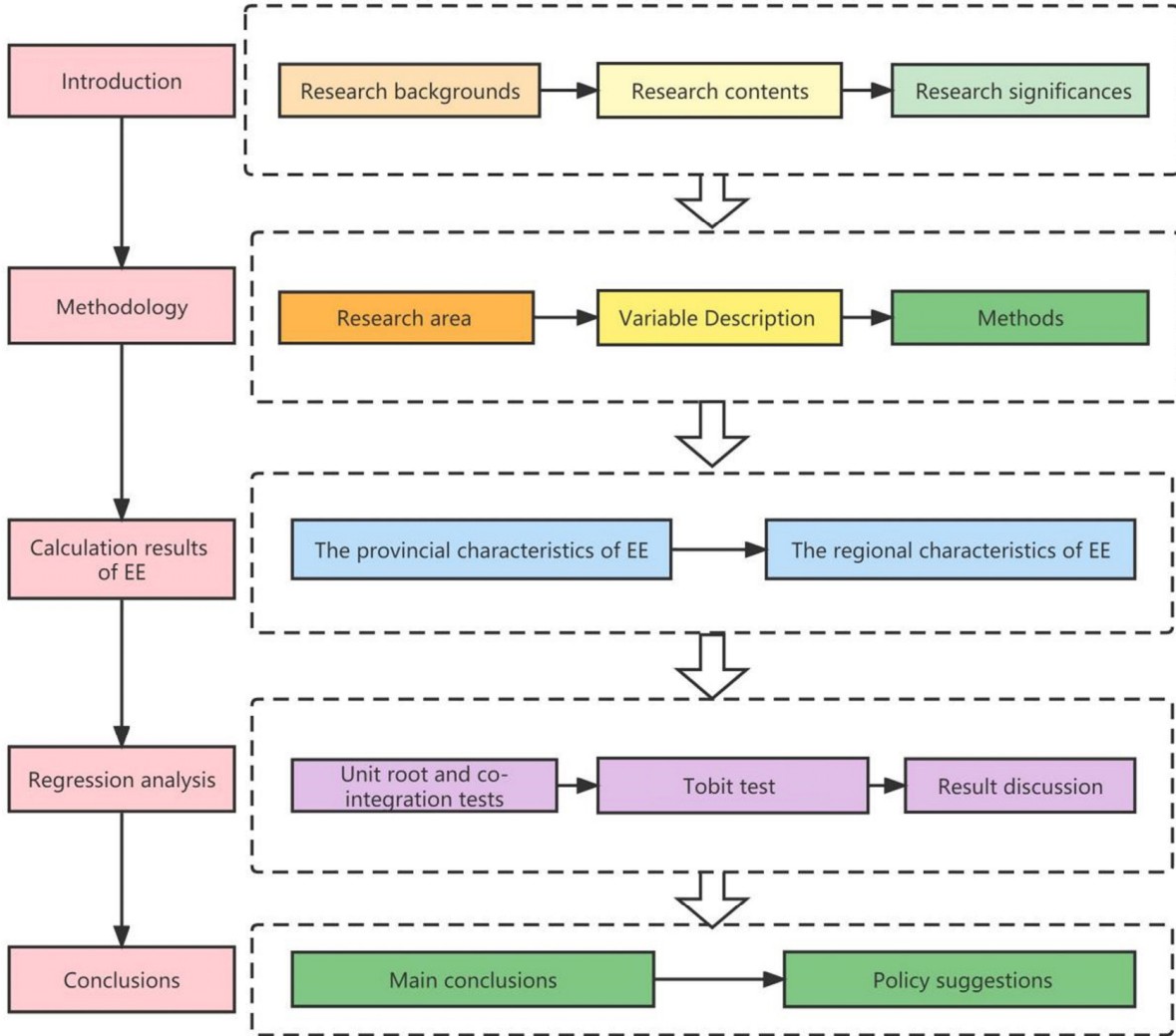

**Figure 1.** The flowchart of the empirical research.

## 2. Data and Methodology

### 2.1. Research Area and Data

Two salient reasons are behind the selection of making China the research background. First, China is the most populous nation in the world and thus engenders extensive demands for both economic growth and energy consumption [6]. As China has become the country with the largest energy consumption and carbon emissions in the world, improving environmental performance is increasingly important for achieving sustainable development in China and beyond.

This paper chose 30 Chinese provinces as the research object, adopting the EBM DEA model with undesirable outputs to measure the eco-efficiency of the Chinese provinces during the period from 2008 to 2019 (due to incomplete data or inconsistent statistical standards of indicators, data from Tibet, Hong Kong, Macao, and Taiwan regions were not included in the analysis). According to the National Bureau of Statistics of China, this paper divided China's economic region into four major parts: the eastern, the central, the western, and the northeast (see Figure 2).

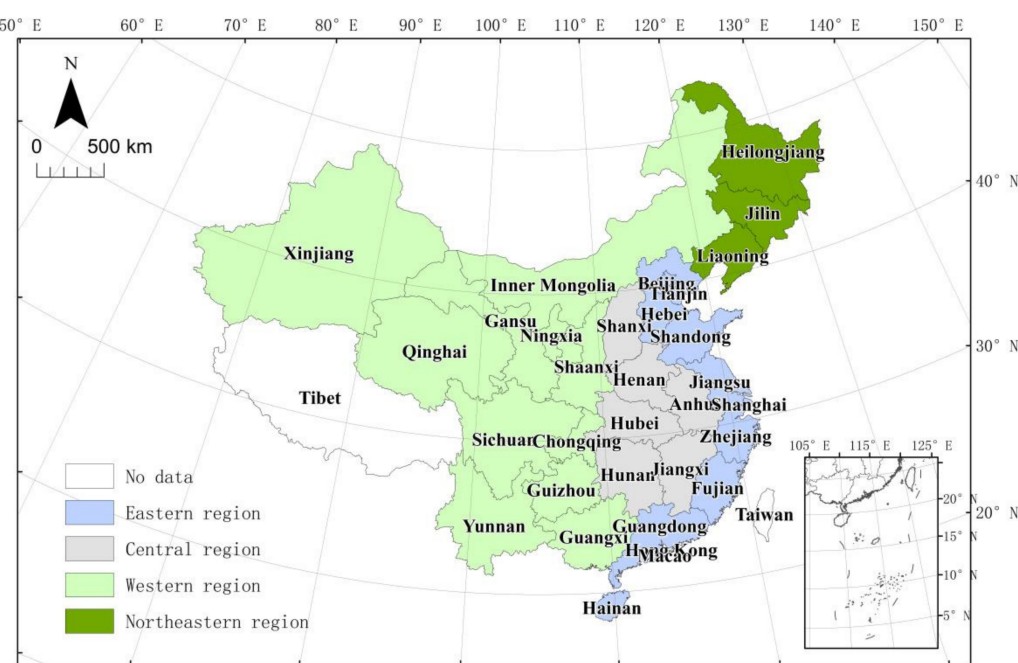

**Figure 2.** The research areas.

### 2.2. Variable Description

2.2.1. Explained Variable

Eco-efficiency. As mentioned above, achieving high *EE* means yielding more economic output through less natural resource consumption and environmental pollution. Therefore, the *EE* evaluation system can be divided into three indicators: the input indicator, the desirable outputs, and the undesirable outputs (Table 1). Input indicators reflect basic economic activities, including labor force, capital stock, construction land, energy consumption and water consumption. Based on Zhang et al. [7], we recalculated capital stocks by employing a perpetual inventory (stock) system. Gross domestic product (GDP) refers to the final value of productive activity in each region, which was selected as desirable output. Other than the desirable outputs, there are also pollutant emissions in the production process, constituting the undesirable outputs [8]. Sulfur dioxide, Carbon dioxide, and Chemical oxygen demand (COD), were selected as the undesirable outputs. In order to eliminate the impact of inflation, all economic data were adjusted to 2008 prices. All input–output indicators were derived from the China Statistical Yearbook (2009–2020) [9], the China Environmental Statistical Yearbook (2009–2020) [10], and the China Emission Accounts and Datasets (https://www.ceads.net/ accessed on 7 March 2022).

**Table 1.** Evaluation index system of Eco-efficiency.

| Primary Indicators | Secondary Indicators | Unit |
|---|---|---|
| Inputs | Capital stock | 100 million Renminbi (RMB) |
| | Employees | 10,000 people |
| | Construction land area | Sq.km |
| | Energy consumption | 10,000 tons of standard coal |
| | Water consumption | $10^8$ L |
| Desired outputs | Gross domestic product | 100 million RMB |
| Undesired outputs | Sulfur dioxide emissions | 10,000 tons |
| | Carbon dioxide emissions | 10,000 tons |
| | Chemical oxygen demand emissions | 10,000 tons |

### 2.2.2. Core Explanatory Variable

The ratio of the urban permanent resident population to the total regional permanent resident population was selected to represent the urbanization level in this study.

The impacts of urbanization on *EE* are controversial, and a series of explanations were put forward. Urbanization refers to a process in which rural people from the countryside move to cities, with their production activities gradually transforming from the first industry to the industrial and the service sectors. In addition, those people's lifestyle also gradually changes to urbanized. However, the development of urban production and lifestyle requires significant products, energy materials, and other natural resource inputs, and results in substantial waste streams and emissions. With the rapid expansion of land use in cities and the growth of population, demands for traffic and energy are increasing [11,12], causing a great deal of environmental pollution and ecological destruction [13]. By contrast, some scholars hold different opinions; Newman, Kenworthy and Ewing believe that there is a negative correlation between transportation emissions and population density. They argued that compact and highly dense urban forms encourage urban dwellers to use public transportation [14–16]. Matsuhashi and Ariga [17] found that bigger population sizes had lower $CO_2$ emission levels of annual per capita passenger cars in Japanese municipalities. Urbanization can generate gathering benefits. Cities, especially the huge ones, will bring an obvious centralization effect and scale effect, which, can promote the development of science and technology, thus cutting down on pollution. The centralized use of energy, information spillover and technological progress brought by urbanization may contribute to the improvement of energy efficiency (characterized by energy intensity), which in turn improves *EE* [18]. Therefore, it is necessary to take the urbanization level as the core explanatory variable in the study.

### 2.2.3. Control Variables

Referring to the existing research, this paper added a group of control variables to the benchmark regression model to mitigate the bias of missing variables as much as possible. Specifically, it included the economic development level (*EDL*), technical progress level, and foreign direct investment (*FDI*) (Table 2).

(1) Economic development level (*EDL*). The theory of the environmental Kuznets curve holds that environmental degradation exacerbates with economic growth at the initial stage before reaching the peak, and then declines when the economy develops to a higher stage [19]. Economic growth offers strong support for technical development and newer, cleaner production, resulting in improved environmental quality [20]. However, according to the rebound effect, economic growth possibly induces more energy consumption and environmental pollution [21]. Therefore, it is essential to examine whether economic growth is a good thing for *EE*.

(2) Technical progress level (*TPL*). The improvement of science and technology can increase economic output and reduce energy consumption [22]. This paper adopted the patent application granted per 100,000 people as the measurement indicator of the technical progress level, and expected that the technical progress level was positively correlated with *EE*.

(3) Foreign direct investment (*FDI*). Previous studies show that *FDI* could lead to knowledge spillovers [23–25] and improved institutional quality in some regions in the host country [26]. However, according to the pollution haven hypothesis (PHH), the emissions of high pollution industries may be transferred to developing countries from developed countries due to weak environmental regulations in developing countries [27]. Hence, this paper selected *FDI* as an important control variable and took the proportion of total *FDI* to GDP as the proxy variable of the opening level. Nevertheless, the direction of *FDI*'s impact on *EE* remains uncertain.

**Table 2.** Influencing factors.

| Explanatory Variable | Definition of the Variables | Pre-Judgment |
|---|---|---|
| Urbanization level (*UL*) | Proportion of the urban resident population to the total population (%) | Unknown |
| Economic development level (*EDL*) | GDP per capita ($10^4$ RMB) | Unknown |
| Technical progress level (*TPL*) | Patent application granted per 100,000 people (item) | Positive |
| Foreign direct investment (*FDI*) | Proportion of the foreign direct investments to GDP (%) | Unknown |

Data source: China Statistical Yearbook (2009–2020) [9], and National Bureau of Statistics of China (2022) [28].

*2.3. Methodology*

2.3.1. The Epsilon-Based Measure (EBM) Model with Undesirable Outputs

In this study, each city is regarded as a decision making unit (DMU) of production, and multiple DMUs. In order to solve the problem of both radial and non-radial DEA models, in 2010, Tone and Tsutsui [29] proposed the epsilon-based measure (EBM) that can combine both radial and non-radial factors. However, the standard EBM model fails to consider the undesirable output factors. To solve these shortcomings, this paper uses a super-efficiency EBM model with undesirable outputs to calculate *EE*, which has two advantages: firstly, combining both radial and non-radial factors; secondly, including undesired environmental output factors [30–32]. The EBM DEA model with undesirable outputs can be represented as follows [33]:

$$\theta^* = \min \left( \frac{\kappa - \varepsilon x \sum_{i=1}^{m} \frac{\omega_i^b s_i^b}{x_{io}}}{\beta + \varepsilon y \sum_{r=1}^{s} \frac{\omega_r^g s_r^b}{y_{ro}} + \varepsilon_b \sum_{p=1}^{q} \frac{\omega_p^b s_p^b}{b_{pk}}} \right)$$

$$s.t \begin{cases} \sum_{j=1}^{n} x_{ij}\lambda_j + s_i^b = \kappa x_{io} & i = 1, 2, \ldots, m \\ \sum_{j=1}^{n} y_{rj}\lambda_j - s_r^g = \beta y_{ro} & r = 1, 2, \ldots, s \\ \sum_{j=1}^{q} b_{pj}\lambda_j + s_p^b\lambda = \beta b_{po} & p = 1, 2, \ldots, q \\ \lambda_j \geq 0, s_i^b \geq 0, s_r^g \geq 0, s_p^b \geq 0 \end{cases} \quad (1)$$

$\theta^*$, $\kappa$, and $\beta$: the technical efficiency the EBM DEA model with undesirable outputs, the radial DEA model, and the non-radial DEA model, respectively.

$n$, $s$, $m$, and $q$: the number of DMUs, the outputs, the inputs, and the undesirable outputs, respectively.

$s_r^g$ and $s_p^b$: the slacks of desired output $r$ and undesired output $p$, respectively.

$\omega_r^g$ and $\omega_p^b$: the desired output weight and the undesired output weight, respectively.

$b_{pk}$: the $p$th undesirable output of the $DMU_k$.

$\varepsilon_y$ and $\varepsilon_b$: the parameters that can combine the radial and non-radial slack.

$\lambda$: the intensity vector.

2.3.2. Tobit

The eco-efficiency value calculated by the EBM-undesirable model is discrete. Generally speaking, since the regression coefficient was calculated, we typically applied the ordinary least squares (OLS), which was discrete in the value of the explained variable. When the regression parameters are estimated, bias and inconsistency may occur. To prevent this, Tobin put forward the intercepted regression model, called the Tobit model, to apply the maximum likelihood method. This method can overcome the shortcomings of OLS. The Tobit method has two advantages: first, the value of the explained variable in the model is discrete; that is, it is observed in a restricted manner. Second, this model applies the maximum likelihood estimation to calculate the regression parameters [34–36].

Therefore, the Tobit method has strong robustness and feasibility. The Tobit model is as follows:

$$Y^* = \beta Xi + ui$$
$$Yi = \begin{cases} Y_i^* & \text{if} \quad Y_i^* > 0 \\ 0 & \text{if} \quad Y_i^* \leq 0 \end{cases} \tag{2}$$

In Equation (2), *i* stands for the ith DMU. $Y^*$ is the latent variable, and $Y_i$ stands for a limited dependent variable. $Y_i$ is the latent variable, $X_i$ is the explanatory variable, $\beta$ represents the correlation coefficient, and u is the random error with the distribution of $N$ $(0, \sigma^2)$. We calculated the regression coefficients by using maximum likelihood estimation in the Stata 12.0 software.

## 3. Calculation Results of *EE*

Significant variations existed in the *EE* of different provinces in China (Figure 3). As Table 3 and Figure 4 show, the *EE*s of Beijing, Shanghai, and Fujian are above 1 throughout the research years, which means that these three provinces realize the coordinated development of economic growth with environmental protection. In the other nine provinces, namely Guangdong, Zhejiang, Jiangsu, Shaanxi, Hunan, Henan, Tianjin, Chongqing, and Hebei, the levels of *EE* are relatively high, with their annual average *EE* higher than the national average. The result indicates high resource conservation and environmental protection in these provinces. The *EE* in the other 18 provinces is far lower than the whole country level. Those provinces should increase their investment in the field of resource utilization, environmental pollution control, and industrial structure adjustment to improve their *EE*.

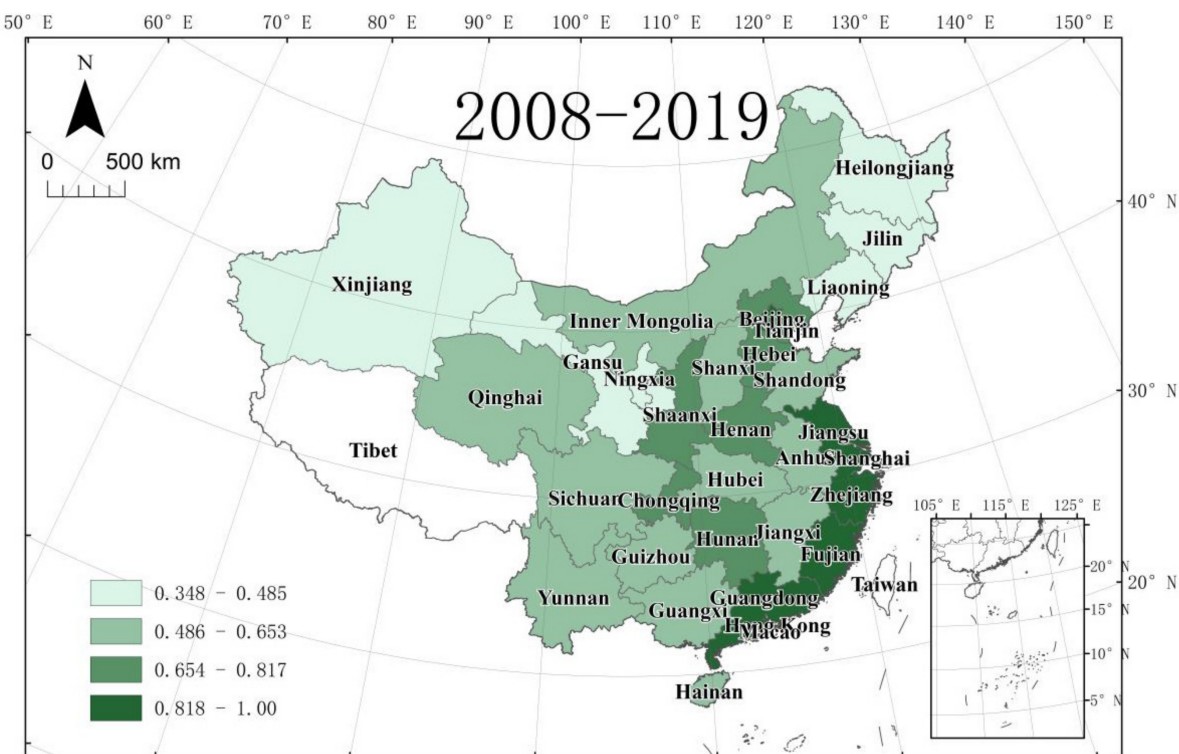

**Figure 3.** Average *EE* values in the 30 Chinese provinces (2008–2019).

**Table 3.** The values of *EE* for 30 provinces in China (2008–2019).

| Provinces | 2008 | 2009 | 2010 | 2011 | 2012 | 2013 | 2014 | 2015 | 2016 | 2017 | 2018 | 2019 | Mean |
|---|---|---|---|---|---|---|---|---|---|---|---|---|---|
| Beijing | 1.000 | 1.000 | 1.000 | 1.000 | 1.000 | 1.000 | 1.000 | 1.000 | 1.000 | 1.000 | 1.000 | 1.000 | 1.000 |
| Tianjin | 0.673 | 0.702 | 0.718 | 0.744 | 0.750 | 1.000 | 1.000 | 0.665 | 0.659 | 0.632 | 0.663 | 0.637 | 0.738 |
| Hebei | 0.699 | 0.703 | 0.694 | 0.701 | 0.684 | 0.610 | 0.649 | 0.634 | 0.635 | 0.656 | 0.749 | 0.754 | 0.681 |
| Shanxi | 0.696 | 0.658 | 0.665 | 0.675 | 0.624 | 0.561 | 0.584 | 0.555 | 0.546 | 0.572 | 0.605 | 0.544 | 0.607 |
| Inner Mongolia | 0.550 | 0.579 | 0.527 | 0.538 | 0.528 | 0.492 | 0.517 | 0.553 | 0.593 | 0.594 | 0.633 | 0.645 | 0.562 |
| Liaoning | 0.507 | 0.521 | 0.516 | 0.520 | 0.511 | 0.467 | 0.492 | 0.480 | 0.441 | 0.445 | 0.456 | 0.465 | 0.485 |
| Jilin | 0.402 | 0.396 | 0.397 | 0.403 | 0.398 | 0.373 | 0.394 | 0.379 | 0.383 | 0.380 | 0.383 | 0.373 | 0.388 |
| Heilongjiang | 0.483 | 0.486 | 0.481 | 0.477 | 0.449 | 0.403 | 0.423 | 0.415 | 0.411 | 0.419 | 0.423 | 0.430 | 0.442 |
| Shanghai | 1.000 | 1.000 | 1.000 | 1.000 | 1.000 | 1.000 | 1.000 | 1.000 | 1.000 | 1.000 | 1.000 | 1.000 | 1.000 |
| Jiangsu | 0.844 | 0.865 | 0.865 | 0.894 | 1.000 | 1.000 | 1.000 | 0.810 | 0.772 | 0.793 | 0.846 | 0.833 | 0.877 |
| Zhejiang | 0.891 | 0.894 | 0.905 | 1.000 | 1.000 | 0.862 | 0.881 | 1.000 | 1.000 | 0.878 | 0.834 | 0.826 | 0.915 |
| Anhui | 0.564 | 0.577 | 0.583 | 0.608 | 0.576 | 0.524 | 0.564 | 0.552 | 0.566 | 0.592 | 0.616 | 0.607 | 0.577 |
| Fujian | 1.000 | 1.000 | 1.000 | 1.000 | 1.000 | 1.000 | 1.000 | 1.000 | 1.000 | 1.000 | 1.000 | 1.000 | 1.000 |
| Jiangxi | 0.633 | 0.653 | 0.650 | 0.672 | 0.647 | 0.575 | 0.624 | 0.602 | 0.611 | 0.612 | 0.631 | 0.630 | 0.628 |
| Shandong | 0.669 | 0.678 | 0.670 | 0.671 | 0.640 | 0.603 | 0.620 | 0.610 | 0.629 | 0.643 | 0.651 | 0.629 | 0.643 |
| Henan | 0.733 | 0.756 | 0.763 | 0.780 | 0.756 | 0.693 | 0.744 | 0.726 | 0.759 | 0.767 | 0.793 | 0.780 | 0.754 |
| Hubei | 0.583 | 0.620 | 0.574 | 0.594 | 0.578 | 0.560 | 0.560 | 0.628 | 0.640 | 0.595 | 0.609 | 0.614 | 0.596 |
| Hunan | 0.677 | 0.661 | 0.690 | 0.727 | 0.746 | 0.690 | 0.759 | 0.809 | 0.841 | 0.829 | 0.825 | 0.844 | 0.758 |
| Guangdong | 1.000 | 1.000 | 1.000 | 1.000 | 1.000 | 1.000 | 1.000 | 1.000 | 0.792 | 0.774 | 0.813 | 0.762 | 0.928 |
| Guangxi | 0.519 | 0.594 | 0.594 | 0.625 | 0.577 | 0.515 | 0.553 | 0.531 | 0.526 | 0.528 | 0.534 | 0.524 | 0.552 |
| Hainan | 0.572 | 0.664 | 0.595 | 0.596 | 0.606 | 0.527 | 0.602 | 0.453 | 0.557 | 0.652 | 0.592 | 0.581 | 0.583 |
| Chongqing | 0.647 | 0.656 | 0.656 | 0.666 | 0.735 | 0.686 | 0.699 | 0.677 | 0.718 | 0.737 | 0.746 | 0.734 | 0.696 |
| Sichuan | 0.680 | 0.703 | 0.700 | 0.716 | 0.680 | 0.599 | 0.636 | 0.632 | 0.615 | 0.614 | 0.632 | 0.629 | 0.653 |
| Guizhou | 0.543 | 0.555 | 0.600 | 0.610 | 0.590 | 0.527 | 0.571 | 0.554 | 0.546 | 0.508 | 0.529 | 0.534 | 0.556 |
| Yunnan | 0.623 | 0.637 | 0.626 | 0.636 | 0.648 | 0.639 | 0.644 | 0.632 | 0.641 | 0.665 | 0.693 | 0.691 | 0.648 |
| Shaanxi | 0.708 | 1.000 | 1.000 | 1.000 | 1.000 | 0.795 | 0.770 | 0.706 | 0.707 | 0.690 | 0.719 | 0.708 | 0.817 |
| Gansu | 0.473 | 0.473 | 0.472 | 0.485 | 0.471 | 0.440 | 0.442 | 0.436 | 0.440 | 0.424 | 0.430 | 0.432 | 0.451 |
| Qinghai | 0.560 | 0.574 | 0.611 | 0.617 | 0.614 | 0.504 | 0.540 | 0.518 | 0.536 | 0.567 | 0.590 | 0.555 | 0.565 |
| Ningxia | 0.352 | 0.356 | 0.383 | 0.374 | 0.359 | 0.330 | 0.343 | 0.333 | 0.340 | 0.341 | 0.332 | 0.330 | 0.348 |
| Xinjiang | 0.463 | 0.461 | 0.461 | 0.457 | 0.432 | 0.385 | 0.405 | 0.390 | 0.393 | 0.400 | 0.407 | 0.439 | 0.424 |
| Eastern region | 0.835 | 0.851 | 0.845 | 0.861 | 0.868 | 0.861 | 0.876 | 0.817 | 0.805 | 0.803 | 0.815 | 0.802 | 0.837 |
| Central region | 0.648 | 0.654 | 0.654 | 0.676 | 0.655 | 0.601 | 0.639 | 0.645 | 0.661 | 0.661 | 0.680 | 0.670 | 0.653 |
| Western region | 0.556 | 0.599 | 0.603 | 0.611 | 0.603 | 0.537 | 0.556 | 0.542 | 0.550 | 0.552 | 0.568 | 0.566 | 0.570 |
| Northeast region | 0.464 | 0.468 | 0.465 | 0.467 | 0.453 | 0.414 | 0.436 | 0.425 | 0.412 | 0.415 | 0.421 | 0.423 | 0.438 |
| Mean | 0.658 | 0.681 | 0.680 | 0.693 | 0.687 | 0.646 | 0.667 | 0.643 | 0.643 | 0.644 | 0.658 | 0.651 | 0.662 |

According to the regional differences, the annual mean *EE* values of Eastern provinces range from 0.8022–0.8755, and those of the central, the western, and the northeastern range from 0.6005–0.6798, 0.5375–0.6113, and 0.4117–0.4677, respectively. Apparently, the comprehensive *EE* level is as the following: the east is the highest, followed by the middle, the west, and the northeast is the lowest. Relying on capital, talent, and technological accumulation, the eastern region has paid attention to the introduction and development of resource conservation and environmental protection technologies, making remarkable achievements in energy-saving and emissions reduction [37,38]. With the national rise of central China in recent years, the central region has made use of regional advantages, developing heavy industry and actively constructing the energy industry base. Since the central location is adjacent to the east with frequent technical communication between them, the *EE* level in the central region is higher than that of the western region. Due to the physical geography, the economic development is chronically slow in west China. The northeastern area belongs to one of the traditional industrial bases in China; thus, the development in industry in the region has characteristics of high consumption of fossil fuel and the difficulties of economic transformation over the past three decades, which affected the regional *EE*.

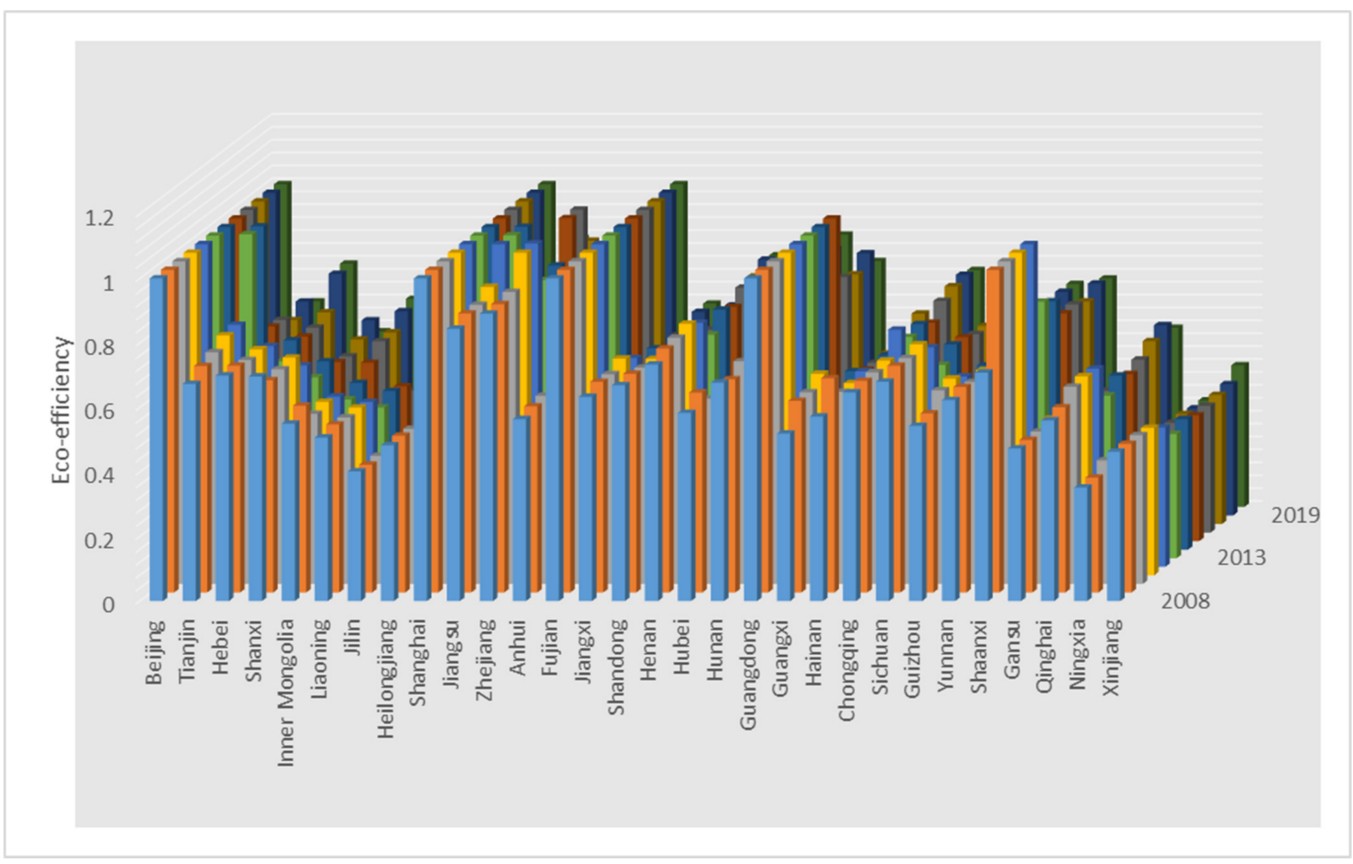

**Figure 4.** The values of *EE* for 30 provinces in China (2008–2019).

## 4. Regression Analysis

### 4.1. Unit Root and Co-Integration Tests

In the positive analysis, considering the characteristics of panel data, the unit root test and co-integration were used to test the variables. This paper applied Levin, Lin, and Chu (LLC), Im, Pesaran, and Shin (IPS), Augmented Dickey Fuller(ADF)-Fisher, and Phillips and Perron(PP)-Fisher tests to examine the stationary properties of all the variables to avoid spurious regression. In Table 4, *UL*, (*UL*) × 2, *DEL*, *TPL*, and *FDI* refer to the urbanization level, the square value of the urbanization level, economic development level, technical progress level, and foreign direct investment. As Table 4 suggests, all the variables are second-order stable, which means that we need to proceed with the co-integration test. We applied the Kao co-integration test to check for a long-run equilibrium relationship among the variables [39]. The results showed that the *t*-Statistic of Augmented Dickey-Fuller was significant at the 1% level, indicating an overwhelming evidence for co-integration between *EE*, *UL*, *UL*× 2, *EDL*, *TPL* and *FDI*. Thus, a co-integration relationship exists (Table 5).

**Table 4.** The results of unit root test.

|  | Levin, Lin, and Chu | Im, Pesaran, and Shin | Augmented Dickey-Fuller-Fisher | Phillips and Perron-Fisher | Conclusion |
|---|---|---|---|---|---|
| $\Delta EE$ | −6.20717 *** | −4.54093 *** | 110.591 *** | 162.666 *** | Stationary |
| $\Delta UL$ | −2.67078 *** | 1.38798 | 82.4110 ** | 72.8774 | No stationary |
| $\Delta UL \times 2$ | 2.95207 | 5.79925 | 69.2863 | 53.6043 | No stationary |
| $\Delta EDL$ | 13.4973 | 17.495 | 35.3251 | 63.3909 | No stationary |
| $\Delta TPL$ | 7.67743 | 11.9316 | 8.2301 | 14.6013 | No stationary |
| $\Delta FDI$ | −4.35353 *** | 0.77825 | 62.7031 | 73.8788 | No stationary |
| $\Delta\Delta EE$ | −21.2810 *** | −15.5620 *** | 277.196 *** | 414.637 *** | Stationary |
| $\Delta\Delta UL$ | −25.6808 *** | −16.3227 *** | 237.875 *** | 226.108 *** | Stationary |
| $\Delta\Delta UL \times 2$ | −36.8898 *** | −19.2961 *** | 250.690 *** | 237.262 *** | Stationary |
| $\Delta\Delta DEL$ | −45.6869 *** | −23.2062 *** | 187.368 *** | 176.307 *** | Stationary |
| $\Delta\Delta TPL$ | −13.9890 *** | −9.34255 *** | 182.766 *** | 216.738 *** | Stationary |
| $\Delta\Delta FDI$ | −14.5359 *** | −7.83367 *** | 169.434 *** | 198.396 *** | Stationary |

Note: ** and *** stand for significance at the 1% and 5%, respectively.

**Table 5.** Co-integration test.

|  | $t$-Statistic | Prob. |
|---|---|---|
| Augmented Dickey-Fuller | −2.785106 | 0.0027 *** |
| Residual variance | 0.002205 | |
| HAC variance | 0.001852 | |

Note: *** stands for significance at the 1% level.

*4.2. Tobit Test*

After the above verifications, we conducted the Tobit regression analysis by applying Stata 15.0. The structural equation was given as follows:

$$EE_{i,t} = \beta_1 UL_{i,t} + \beta_2 UL \times 2_{i,t} + \beta_3 DEL_{i,t} + \beta_4 TPL_{i,t} + \beta_5 FDI_{i,t} + \varepsilon_{i,t} \tag{3}$$

In Equation (3), *UL*, *(UL)* × 2, *DEL*, *TPL*, and *FDI* refer to the urbanization level, the square value of the urbanization level, economic development level, technical progress level, and foreign direct investment, respectively; $\varepsilon$ is the stochastic disturbance item.

The parameter estimation results by the Tobit model are listed in Table 6. The results show that each variable has a different influence on *EE*. The results show that the urbanization level, economic development level, technical progress level, and foreign direct investments have passed the significance test at the 1% level, meaning that there is a 99% probability that these variables have a significant influence on the *EE*. The detailed analysis of each explanatory variable is as follows.

**Table 6.** The regression results of Tobit model.

|  | Coef. | St.Err. | $t$-Value | $p$-Value |
|---|---|---|---|---|
| UL | −2.713 *** | 0.533 | −5.09 | 0.00 |
| UL × 2 | 1.853 *** | 0.439 | 4.22 | 0.00 |
| DEL | 0.063 *** | 0.016 | 3.93 | 0.00 |
| TPL | 1.907 *** | 0.531 | 3.59 | 0.00 |
| FDI | 1.243 *** | 0.153 | 8.12 | 0.00 |
| Constant | −2.713 *** | 0.533 | −5.09 | 0.00 |

Note:*** stands for significance at the 1%.

The parameter estimation value of the urbanization level, and the urbanization level × 2 registers −2.713, and 1.853, respectively, suggesting that the functional relationship between urbanization and *EE* may be inverted U-shaped. The cause behind this phenomenon may be China's pursuit of a proactive fiscal policy and an expanded urbanization strategy in response to the 2008 outbreak of the world economic crisis, which

presupposed massive infrastructure and housing and accelerated overall energy consumption. At the same time, massive population migration from rural areas to urban cities generated energy consumption and pollution emissions, which destroyed the construction of the ecological economy, and the *EE* declined. In the current stage, urbanization has negative impacts on *EE*. However, this relationship would not last forever. Since the 18th CPC National Congress in 2012, China has accelerated the construction of Ecological Civilization and has made a pledge to transform urbanization and economic development models. Therefore, a new type of urbanization strategy has been implemented. On the other hand, the improvement of urbanization can bring the cluster effect of human capital and the spillover effect of advanced production technologies to cities. As a result, the process of urbanization not only brings about enormous economic growth, but also the development and application of clean production technology; therefore, the impact of urbanization on *EE* would transform into a positive correlation for the future.

As for the control variables, the regression coefficient of the economic development level shows a significantly positive correlation. At present, China is the second largest economy in the world, which can provide substantial financial support for the transformation and the upgrading of industries. The advantages guarantee the development of production technology, promising a positive impact on the *EE* in China. The estimated coefficients of technological progress level are positive, exceeding 1%. This result further verifies that the "innovation drive" is the key to China's economic transformation, and technological progress can bring about the improvement of production, environmental protection technology, and efficiency, which is the key to creating a good development environment for new urbanization. Foreign direct investments play a significant role in promoting *EE*. At one level, the increase in foreign direct investment promotes growth and creates output value and more regional job opportunities. In addition, technology spillovers and demonstration effects ultimately improve *EE* [40].

## 5. Conclusions and Policy Suggestions

The rapid urbanization and continuous economic growth in China are accompanied by new environmental issues [41]. This paper used the panel data of 30 provinces in China from 2008 to 2019, and comprehensively used the DEA method, the Tobit method, and ArcGIS Geographic Information analytical methods to analyze the impact of urbanization on *EE*. Major findings are summarized as follows: firstly, the eastern region has the highest rate of *EE*, followed by the central and western regions, and the northeast region remains the lowest. Secondly, the *EE* of Beijing, Shanghai, and Fujian were at the production frontier surface with a high level during the study period. The *EE* of Gansu, Ningxia, and Xinjiang were generally at a low level. A regional mechanism of energy-saving and emissions reduction should be built to reduce regional differences [42]. Thirdly, the effects of urbanization on *EE* in China present a U-shaped relationship, having a negative relation first and then reversing to a positive one. At present, the process of urbanization shows negative impacts on the *EE*, while the turning point is yet to come. As for the control variables, the economic development level, technological progress, and foreign direct investments have positive impacts on *EE*.

Based on the results of the empirical study, several policy suggestions may be proposed: (1) the local governments should take diversified roads of high-quality urbanization according to the local conditions. The eastern region has advanced production technology, management, and rich capital, so it should adopt the most radical model of a new-type urbanization strategy. The central and western regions are extremely rich in various natural and tourism resources, such as minerals, energy, water power, wind power, and geothermal heat, etc. With the implementation of new-type urbanization, they can cultivate an economy with characteristics and make the best use of the comparative advantages. (2) The process of urbanization should take account of the effective utilization of resources and energy. The government should accelerate reforms of natural resource commodity pricing based on the degree of scarcity, and encourage and support the development of a recycling economy

to promote renewable resources recovery and utilization. The government should also introduce tax preferences to encourage firms to upgrade their production technology and apply clean production technologies. (3) It is also wise to strengthen environmental protection regulations when attracting foreign investment to avoid importing high pollution and high energy consumption industries. In the future, to boost economic development, high-tech industries, environmentally friendly industries, and producer services industries will undoubtedly be welcomed in China.

**Author Contributions:** X.Y.: conceptualization, methodology, software, validation, formal analysis. Y.N.: writing—original draft preparation, writing—review and editing, visualization, supervision, project administration, funding acquisition. L.Z.: investigation, resources, data curation, writing—original draft preparation, writing—review and editing. C.L.: data curation. T.Y.: investigation, resources, data curation, writing—review and editing. All authors have read and agreed to the published version of the manuscript.

**Funding:** This research was funded by the Carbon Neutralization Promotion Fund of the China Green Carbon Foundation.

**Institutional Review Board Statement:** Not applicable.

**Informed Consent Statement:** Not applicable.

**Data Availability Statement:** Data were obtained from the China official national statistical database and the China Emission Accounts and Datasets.

**Conflicts of Interest:** The authors declare no conflict of interest.

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
