# Peer review of "Exploring the Impacts of Urbanization on Eco-Efficiency in China"

_land, doi:10.3390/land12030687_

Round 1

Reviewer 1 Report

The issue taken up in this article is important and up-to-date. And that makes it interesting. The authors raised the problem of the impact of urbanization on eco-efficiency, and this is very important due to the high dynamics of climate change. The choice of China as the subject of research is also very good, because its impact on the global climate is huge due to China's size and its characteristic demographic, social and economic changes. This issue is quite often taken up by Chinese researchers, which I assess very positively.

The title of this article is well worded and describes its content well.

This article is a very good description of the complex problem of the impact of urbanization on China's eco-efficiency. Although the concept of urbanization and the concept of eco-efficiency are not simple concepts, in the methodological part of this work they have been clearly defined and broken down into important components. Therefore, I highly appreciate the methodological part of this work. However, I believe that in section 2.2.3., subsection (1) (lines 150-151) the assumption should be strengthened to make it comparable to the other assumptions. Now it is: "This paper assumed that the economic development level has an impact on EE". Or maybe this proposal would be better: "This paper assumed that high the economic development level has an impact on high EE"? Please think it over.

Discussion of the obtained results and their discussion were carried out correctly. However, the use of abbreviations in the text of the work requires some ordering (detailed comments are included in the manuscript in PDF). First introduce the important terms by giving their full names and then their abbreviations. After this procedure, you can use abbreviations in the content of the work.

I positively evaluate the conclusions of the conducted analytical procedure. I also very positively assess the recommendations for future political action. Basing political actions on objective recommendations is rational and is characteristic of wise leaders.

Author Response

Dear Editor,

Thank you very much for your great comments and suggestions on our paper. We have modified the manuscript accordingly.

(1) In section 2.2.3., subsection (1) (lines 150-151) the assumption should be strengthened to make it comparable to the other assumptions. Now it is: "This paper assumed that the economic development level has an impact on EE". Or maybe this proposal would be better: "This paper assumed that high the economic development level has an impact on high EE"? Please think it over.

Response: The authors have added more theoretical assumptions in lines 157 to 164.

  • Discussion of the obtained results and their discussion were carried out correctly. However, the use of abbreviations in the text of the work requires some ordering (detailed comments are included in the manuscript in PDF). First introduce the important terms by giving their full names and then their abbreviations. After this procedure, you can use abbreviations in the content of the work.s.

Response: The author has made more discussions of the obtained results in line 356 to 366. The abbreviations have been first mentioned when they are used in the passage.

The manuscript has been resubmitted to your journal now.

Regards

Yang Nie

Reviewer 2 Report

1- What's your proof for the first sentence of the abstract section?

2- What does the EBM model stand for?

3- Based on which criteria, 30 provinces are selected?

4- Why you used the 2008 to 2019 time interval? What about 2019 to now?

5- What do you mean by undesirable outputs?

6- The abstract is more similar to the conclusion section. It must be revised.

7- You should present some quantitative results in the abstract section.

8- You can use more suitable keywords.

9- There are some grammatical mistakes in your paper. Please ask a native to revise your paper.

10- In some parts of your paper, you need to use more references.

11- The abbreviations must be mentioned only the first time when they are used in the passage.

12- The flowchart can be present in a more user-friendly type.

13- Please add the geographical coordinate to Fig. 2 and follow the cartographical procedure.

14- You present the research area, but there is no data section in the data and methodology section.

15- What does RMB stand for?

16- Can you present the results of table 3 in a figure? For example, by taking advantage of the Space-Time cube?

17- The titles of the Table. 4 must be justified.

Author Response

Dear Editor,

Thank you very much for your great comments and suggestions on our paper. We have modified the manuscript accordingly.

(1) What's your proof for the first sentence of the abstract section?”

Response: The authors have changed the first sentence of the abstract section

(2) What does the EBM model stand for?

Response: The authors have given the full name of the EBM model in lines 75, and have shown the specific introduction in 191 to 199.

(3) Based on which criteria, 30 provinces are selected?.

Response: The most important reason is that there is no municipal level data of energy consumption data in China. Due to incomplete data or inconsistent statistical standards of indicators, data from Tibet, Hong Kong, Macao, and Taiwan regions were not included in the analysis. Therefore, 30 provinces are selected in this study.

(4) Why you used the 2008 to 2019 time interval? What about 2019 to now?.

Response: There are some data from 2019 to 2022 and before 2008 that we could not collect that. Therefore, on the base of the accessibility of the statistics, we have selected the 2008 to 2019 time interval.

(5)What do you mean by undesirable outputs?

Response: The authors have shown the specific introduction of undesirable outputs in 117 to 118

(6)The abstract is more similar to the conclusion section. It must be revised.

Response: The authors have revised the abstract.

(7) You should present some quantitative results in the abstract section.

Response: The authors have added the quantitative results in the abstract section.

(8)You can use more suitable keywords.

Response: The authors have added new keywords.

(9)Here are some grammatical mistakes in your paper. Please ask a native to revise your paper.

Response: The manuscript has been checked and modified by a native

(10) In some parts of your paper, you need to use more references.

Response: The authors have added more references.

(11)The abbreviations must be mentioned only the first time when they are used in the passage.

Response: The abbreviations have been first mentioned when they are used in the passage.

(12)The flowchart can be present in a more user-friendly type

Response: The flowchart has been modified into a more user-friendly type

(13)Please add the geographical coordinate to Fig. 2 and follow the cartographical procedure.

Response: The authors have added the geographical coordinate to Fig. 2 and followed the cartographical procedure.

(14)You present the research area, but there is no data section in the data and methodology section..

Response: The authors have added the data section in the data and methodology section.

(15)What does RMB stand for?

Response: RMB stands for Renminbi.

(16)Can you present the results of table 3 in a figure? For example, by taking advantage of the Space-Time cube?

Response: The authors have added figure 4 as the results of table 3 

(17)The titles of the Table. 4 must be justified.?.

Response: The authors have added the test results of stationary among variables

The manuscript has been resubmitted to your journal now.

Regards

Yang Nie

Round 2

Reviewer 2 Report

This paper can be accepted in its present form.